**Data Availability Statement:** All relevant data are available in figshare (https://figshare.com/), DOI: 10.6084/m9.figshare.13514146.

# Protective role of resolvin D1, a pro-resolving lipid mediator, in nonsteroidal anti-inflammatory drug-induced small intestinal damage

**Takuya Kuzumoto[1], Tetsuya Tanigawa[1,2]\*, Akira Higashimori[1], Hiroyuki Kitamura[1], Yuji Nadatani[1], Koji Otani[1], Shusei Fukunaga[1], Shuhei Hosomi[1], Fumio Tanaka[1], Noriko Kamata[1], Yasuaki Nagami[1], Koichi Taira[1], Toshio Watanabe[1], Yasuhiro Fujiwara[1]**

1 Department of Gastroenterology, Osaka City University Graduate School of Medicine, Osaka, Japan,
2 Department of Gastroenterology, Osaka City Juso Hospital, Osaka, Japan

\* ttanigawa@med.osaka-cu.ac.jp

## Abstract

Resolvin D1, a specialized pro-resolving lipid mediator produced from docosahexaenoic acid by 15- and 5-lipoxygenase, exerts anti-inflammatory effects driving to the resolution of inflammation. The present study aimed to elucidate its role in small intestinal damage induced by nonsteroidal anti-inflammatory drug (NSAID). Indomethacin was administered orally to C57BL/6J male mice, which were sacrificed 24 h later to collect small intestine specimens. Before administration of indomethacin, mice were subjected to intraperitoneal treatment with resolvin D1 or oral administration of baicalein, a 15-lipoxygenase inhibitor. Small intestinal damage induced by indomethacin was attenuated by pretreatment with resolvin D1. Furthermore, resolvin D1 reduced the gene expression levels of interleukin-1β, tumor necrosis factor-α, and CXCL1/keratinocyte chemoattractant. Conversely, the inhibition of 15-lipoxygenase activity by baicalein increased the expression of genes coding for these inflammatory cytokines and chemokine, leading to exacerbated small intestinal damage, and reduced the concentration of resolvin D1 in the small intestinal tissue. Exogenous treatment with resolvin D1 negated the deleterious effect of baicalein. 15-lipoxygenase was mainly expressed in the epithelium and inflammatory cells of the small intestine, and its gene and protein expression was not affected by the administration of indomethacin. Inhibition of the resolvin D1 receptor, lipoxin A4 receptor /formyl peptide receptor 2, by its specific inhibitors Boc-1 and WRW4 aggravated indomethacin-induced small intestinal damage. Collectively, these results indicate that resolvin D1 produced by 15-lipoxygenase contributes to mucoprotection against NSAID-induced small intestinal damage through its anti-inflammatory effect.

**Funding:** The authors received no specific funding for this work.

**Competing interests:** The authors have declared that no competing interests exist.

## Introduction

Inflammation can be resolved through elimination of pathogens and foreign harmful substances and reduction of inflammatory mediators and cytokines via non-specific metabolic process. However, recent accumulating evidence has revealed that specialized pro-resolving lipid mediators (SPMs) play key roles in regulating the resolution of acute inflammation [1, 2]. SPMs include lipoxins, which are derived from the ω-6 polyunsaturated fatty acid (PUFA) arachidonic acid, and resolvins and protectins, which are derived from the ω-3 PUFAs docosahexaenoic acid (DHA) and eicosapentaenoic acid (EPA). Resolvins are categorized into D-series resolvins synthesized from DHA and E-series resolvins synthesized from EPA [3, 4]. In mice, DHA is converted to 17(S)-hydroxy-DHA via 12/15-lipoxygenase, an ortholog of the human 15-lipoxygenase, which is further converted to endogenous resolvin D by 5-lipoxygenase [5, 6].

Resolvin D1 (7S,8R,17S-trihydroxy-4Z,9E,11E,13Z,15E,19Z-DHA) is one of the most studied resolvins. Resolvin D1 can exert anti-inflammatory effects, such as inhibition of neutrophil recruitment and suppression of production of inflammatory cytokines, via two G protein-coupled receptors, lipoxin A4 receptor (ALX)/formyl peptide receptor 2 (FPR2), and G protein-coupled receptor 32 (GPR32) [7–9]. Moreover, resolvin D1 has been shown to ameliorate various acute inflammatory diseases in animal models, including lung injury, kidney injury, pancreatitis, hepatitis, and peritonitis [10–14].

Nonsteroidal anti-inflammatory drugs (NSAIDs) are one class of frequently prescribed drugs with antipyretic analgesic properties. However, these drugs induce gastrointestinal mucosal damage, mainly in the upper gastrointestinal tract, as an adverse effect. Furthermore, recent technological advances in small intestinal endoscopy have revealed that NSAIDs frequently induce mucosal damage also in the small intestine [15, 16]. In particular, we previously reported mild mucosal damage in 25% and severe mucosal damage in 27.8% of patients with rheumatoid arthritis who took NSAID orally for more than 3 months, as revealed by small intestinal capsule endoscopy [17]. NSAID-induced small intestinal damage occurs as a result of prostaglandin deficiency due to NSAID-dependent inhibition of cyclooxygenase activity and mitochondrial dysfunction [18]. However, the significance of other lipid mediators, including resolvin D1, in NSAID-induced small intestinal damage remains unknown. In this study, we investigated the role of resolvin D1 in NSAID-induced small intestinal damage using an experimental animal model.

## Materials and methods

### Animals

Male C57BL/6 mice, 8–10 weeks old, were purchased from CLEA Japan Inc. (Tokyo, Japan). All mice were kept in polycarbonate cages, maintained under a 12-h light/12-h dark cycle, and allowed free access to standard rodent diet (CE-2; CLEA Japan Inc.) and water. When performing invasive procedures, including tail vein injection and euthanasia, mice were anesthetized via inhalation of isoflurane. Euthanasia was performed by cervical dislocation under deep anesthesia with isoflurane. All procedures were approved by the Institutional Animal Care and Use Committee of the Osaka City University Graduate School of Medicine (Approval No. 18073).

### Induction of NSAID-induced small intestinal damage

To induce small intestinal damage, 10 mg/kg body weight (BW) of indomethacin (FUJIFILM Wako Pure Chemical Corporation, Osaka, Japan) suspended in a 0.5% carboxymethylcellulose (CMC) solution was orally administered to non-fasted mice, which were sacrificed 0, 3, 6, 12, or

24 h after administration. In some experiments, 60 mg/kg BW of diclofenac with vehicle (0.5% CMC) was orally administered instead of indomethacin. The assessment of animal health and well-being was done 24 h before indomethacin administration and 3,6, and 24 h after indometha-cin administration. At each time point, the critical problem about general status of the animal health was not observed in all of the animals. For evaluation of macroscopic intestinal damage, 1% Evans blue was injected intravenously 30 min before sacrifice to delineate the lesions. The entire small intestine was collected and cut open along the antimesenteric side. The areas of lesions delin-eated with Evans blue were measured macroscopically, summed for each small intestine, and expressed in terms of lesion index. The lesions were measured by two investigators (T.K. and T.T.) in a blinded manner. For histological evaluation, each of the small intestinal tissue samples that exhibited typical mucosal damage was fixed with 10% buffered formalin, and 4 μm-thick tissue sections were mounted on glass slides and subjected to hematoxylin and eosin (H&E) staining.

## Histological evaluation of small intestinal damage

Tissue sections stained with H&E were viewed under high power using a white-light micro-scope. For each mouse, at least 10 random villi in the damaged areas were scored independ-ently by two investigators (H.K. and T.T.) in a masked fashion. For evaluation, a modified histological scoring system was used [19]. Histological scores ranged from 0 to 13 and were divided into the following six categories: epithelium (0 = normal, 1 = flattened, 2 = loss of epi-thelial continuity, 3 = severe denudation), villus shape (0 = normal, 1 = short and rounded, 2 = extremely short and thick), villus tip (0 = normal, 1 = damaged, 2 = severely damaged), stroma (0 = normal, 1 = slightly retracted, 2 = severely retracted), inflammation (0 = no infil-tration, 1 = mild infiltration, 2 = severe infiltration), and crypt status (0 = normal, 1 = mild crypt loss, 2 = severe crypt loss).

## Reagents for analysis of the roles of resolvin D1 and 12/15-lipoxygenase

For the experiments to evaluate the effect of exogenous resolvin D1 on NSAID-induced small intestinal damage, 10 μg/kg BW of resolvin D1 (Cayman Chemical Company, Ann Arbor, MI, USA) in phosphate-buffered saline (PBS) was intraperitoneally administered 3 and 24 h before administration of indomethacin. To examine the effect of endogenous resolvin D1, 200 mg/kg BW of baicalein (Tokyo Chemical Industry, Tokyo, Japan), the 12/15-lipoxygenase inhibitor, in PBS was administered 3 and 24 h before administration of indomethacin. Boc-1 (Bachem AG, Bubendorf, Switzerland) and WRW4 (R&D Systems, Minneapolis, MN, USA) were used as inhibitors of the resolvin D1 receptor ALX/FPR2. Boc-1 (5 mg/kg BW) or WRW4 (1 mg/kg BW) was intraperitoneally administered 3 and 24 h before administration of indomethacin.

## Measurement of resolvin D1 concentration in small intestinal tissue

Small intestinal tissue samples were weighed and placed in tubes containing cold 100% metha-nol. Samples were then homogenized on ice and centrifuged at 20,000 $\times g$ for 20 min at 4˚C. After the supernatant was evaporated with $N_2$ gas on ice, assay buffer was added to the residue. The concentrations of resolvin D1 were measured with an enzyme immunoassay (Resolvin D1 EIA kit; Cayman Chemical Company) according to the manufacturer's protocol.

## Quantitative real-time reverse transcription polymerase chain reaction (RT-PCR) analysis

Total RNA was extracted from intestinal tissue using ISOGEN II (Nippon Gene Co., Ltd., Tokyo, Japan). Complementary DNA was synthesized from RNA using the High Capacity

RNA-to-cDNA Kit (Thermo Fisher Scientific Inc., Waltham, MA, USA). Quantitative real-time RT-PCR analyses were performed using an Applied Biosystems 7500 Fast Real-Time PCR system with TaqMan Fast Advanced Master Mix (Thermo Fisher Scientific Inc.). The thermal cycling conditions consisted of 45 cycles of 95˚C for 15 s and 60˚C for 1 min. The mRNA levels of interleukin 1β (*Il1b*), tumor necrosis factor α (*Tnfa*), the mouse *IL-8* homolog CXCL1/keratinocyte chemoattractant (*Cxcl1*), and 12/15-lipoxygenase (*Alox15*) in small intestinal tissue were quantified by quantitative real-time RT-PCR and normalized to the gene expression levels of glyceraldehyde-3-phosphate dehydrogenase (*Gapdh*; Thermo Fisher Scientific Inc.). The mRNA levels were expressed as ratios of the mean values for the control group of mice. The sequences of PCR primers and TaqMan probes for *Il1b*, *Tnfa*, *Cxcl1*, and *Alox15* are shown in S1 Table. RT-PCR analysis was performed at the Research Support Platform of Osaka City University Graduate School of Medicine.

## Western blotting analysis

Intestinal tissue was homogenized on ice in a solution containing TNE buffer (50 mM Tris-HCl pH 7.8, 1% NP-40, 150 mM NaCl, 10 mM ethylenediaminetetraacetic acid), PhosSTOP (one tablet per 10 mL of TNE buffer; Roche Applied Science, Indianapolis, IN, USA), and cOmplete™ Mini Protease Inhibitor Cocktail (one tablet per 10 mL of TNE buffer; Thermo Fisher Scientific Inc.). The supernatant was collected after centrifugation at 10,000 ×*g* for 10 min at 4˚C. Protein concentrations were measured using a Pierce BCA Protein Assay Kit (Thermo Fisher Scientific Inc.). Proteins were denatured with sample buffer solution (FUJI-FILM Wako Pure Chemical Corporation) at 95˚C for 5 min, subjected to 15% SDS-polyacrylamide gel electrophoresis, and transferred to an Immune-Blot polyvinylidene difluoride membrane (Bio-Rad Laboratories, Hercules, CA, USA). DynaMarker Protein MultiColor Stable (BioDynamics Laboratory, Tokyo, Japan) was used as a protein ladder marker. The membranes were blocked with blocking buffer (5% skim milk and Tris-buffered saline with 0.1% Tween 20 [TBS-T]) for 1 h at room temperature and then incubated with rabbit polyclonal antibodies against 12/15-lipoxygenase (#Bs-6505R; Bioss Antibodies, Woburn, MA, USA) and ALX/FPR2 (#NBP1-90180; Novus Biologicals, Littleton, CO, USA) overnight at 4˚C. After washing with TBS-T, the membranes were incubated with anti-rabbit IgG-HRP (#NA934-1ML; GE Healthcare, Little Chalfont, UK) in blocking buffer for 1 h at room temperature. The antigen-antibody complexes were visualized by enhanced chemiluminescence with ECL Prime Western Blotting Detection Reagent (GE Healthcare) and detected using an ImageQuant LAS 4000 Mini instrument (GE Healthcare).

## Immunohistochemistry

For immunohistochemical staining, tissue samples were fixed with 4% paraformaldehyde in 0.1 μM phosphate buffer (pH 7.4) and embedded in Tissue-Tek® O.C.T. Compound (Sakura Finetek Japan, Tokyo, Japan). Six-micrometer-thick cryostat sections were placed on silanized slides (Dako, Tokyo, Japan). Tissue samples were then incubated overnight at 4˚C with primary antibodies, including rabbit polyclonal anti-12/15-lipoxygenase (diluted 1:800, #Bs-6505R, Bioss Antibodies), rabbit polyclonal anti-FPR2 (diluted 1:300, #NBP1-90180, Novus Biologicals), and rat monoclonal anti-F4/80, targeting a marker for mature macrophages (diluted 1:500, #MCA497GA, AbD Serotec, Oxford, UK). For 12/15-lipoxygenase detection, the sections were incubated for 30 min with the secondary antibody EnVision+ System-HRP Labeled Polymer Anti-Rabbit (Dako), treated for 1 min with the DAB substrate of a Liquid DAB+ Substrate Chromogen System (Dako), and counterstained with Mayer's hematoxylin. For FPR2 and F4/80 detection, the sections were incubated with the corresponding secondary

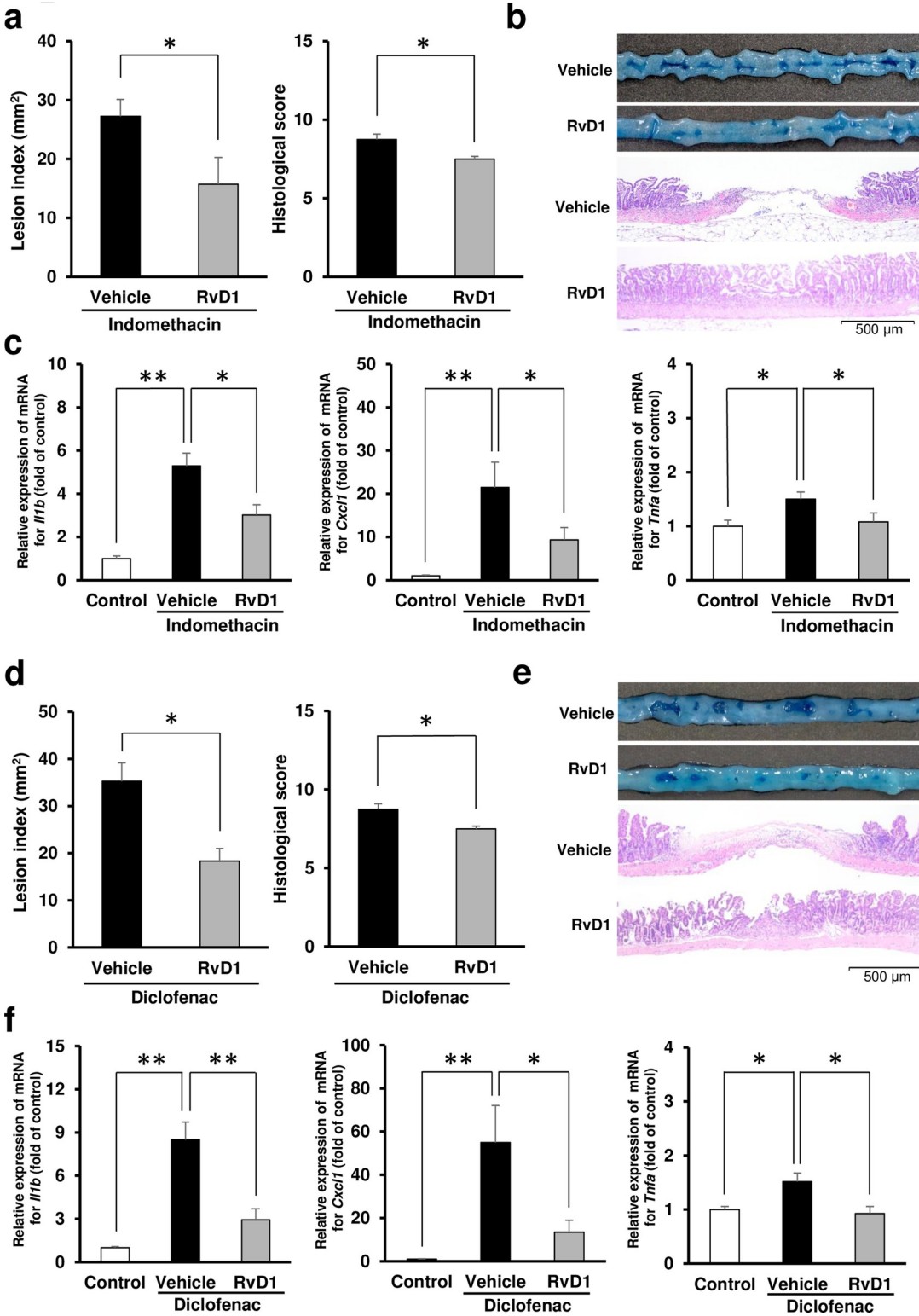

**Fig 1. Effects of exogenous resolvin D1 on NSAID-induced small intestinal damage.** (a) Lesion index and histological score of the small intestines 24 h after administration of indomethacin (10 mg/kg BW) in mice pretreated with vehicle or resolvin D1 (RvD1: 10 µg/kg BW). $N$ = 7–8. (b) Representative macroscopic and microscopic images of small intestine 24 h after administration of indomethacin in mice pre-administered with RvD1 (10 µg/kg BW) or vehicle. Bars in histological images: 500 µm. (c) The gene expression levels of interleukin-1β (*Il1b*), CXCL1/keratinocyte chemoattractant (*Cxcl1*), and tumor

necrosis factor-α (*Tnfa*) 6 h after administration of indomethacin in mice pretreated with vehicle or RvD1 (10 μg/kg BW). The mRNA levels were determined by quantitative reverse transcription polymerase chain reaction and expressed as ratios relative to the mean value for small intestinal tissue from non-treated control mice. *N* = 6–8. (d) Lesion index and histological score of small intestines 24 h after administration of diclofenac (60 mg/kg BW) in mice pretreated with vehicle or RvD1 (10 μg/kg BW). (e) Representative macroscopic and microscopic images of small intestine 24 h after administration of diclofenac in mice pre-administered with RvD1 (10 μg/kg BW) or vehicle. (f) The gene expression levels of *Il1b*, *Cxcl1*, and *Tnfa* 6 h after administration of diclofenac with vehicle or RvD1 (10 μg/kg BW). *N* = 6–8. All data are from a single experiment, representative of at least two independent experiments and expressed as mean ± standard error of mean. * *p* < 0.05, ** *p* < 0.01.

fluorescent dye-conjugated antibodies (Invitrogen, Carlsbad, CA, USA) for 2 h and then 4',6-diamidino-2-phenylindole (DAPI) Fluoromount-G (Southern Biotech, Birmingham, AL, USA) was added for visualization of the nucleus. The sections were examined using a confocal microscope equipped with argon and argon–krypton laser sources (Keyence, Osaka, Japan).

## Statistical analysis

Values are expressed as the mean ± standard error of the mean. One-way analysis of variance was used to test for significant differences among the means of the treatment groups, and results were analyzed using a protected Fisher's least significant difference test. Differences with *p* values < 0.05 were considered statistically significant. All statistical analyses were performed using EZR v.1.40 (Saitama Medical Center, Jichi Medical University, Saitama, Japan), which is a modified graphical user interface for R (The R Foundation for Statistical Computing, Vienna, Austria) designed to add statistical functions frequently used in biostatistics [20].

## Results

### Effects of exogenous resolvin D1 on NSAIDs-induced small intestinal damage

Pretreatment with resolvin D1 attenuated small intestinal damage induced by indomethacin to 58% in terms of lesion index and 86% in terms of histological score (Fig 1A and 1B). Indomethacin increased the gene expression levels of *Il1b*, *Tnfa*, and *Cxcl1*, while resolvin D1 reduced the gene expression levels of *Il1b*, *Tnfa*, and *Cxcl1* to 57%, 72%, and 44%, respectively, in comparison with those of the indomethacin-administered vehicle-treated group (Fig 1C). Similar to indomethacin-dependent damage, diclofenac-induced small intestinal damage was also attenuated by pretreatment with resolvin D1. Indeed, resolvin D1 reduced the lesion index to 52% and the histological score to 84% (Fig 1D and 1E). Moreover, the gene expression levels of *Il1b*, *Tnfa*, and *Cxcl1* were reduced to 35%, 61%, and 24%, respectively, in comparison with those of the diclofenac-administered vehicle-treated group (Fig 1F).

### Dynamics of 12/15-lipoxygenase expression in small intestinal tissue

Immunoreactivity against 12/15-lipoxygenase was observed in intestinal epithelial cells as well as in inflammatory cells of the lamina propria (Fig 2A). The mRNA levels of *Il1b*, *Tnfa*, and *Cxcl1* in the small intestine peaked 6 to 12 h after indomethacin administration. In contrast, the mRNA levels of 12/15-lipoxygenase (*Alox15*) in the small intestine did not change over time (Fig 2B). Similar to the mRNA profile, the protein concentration of 12/15-lipoxygenase in the small intestine was almost constant after the administration of indomethacin (Fig 2C and 2D). Consistently, there was no difference in concentration of resolvin D1 in the small intestine between the vehicle-administered control group and indomethacin-administered group (Fig 2E).

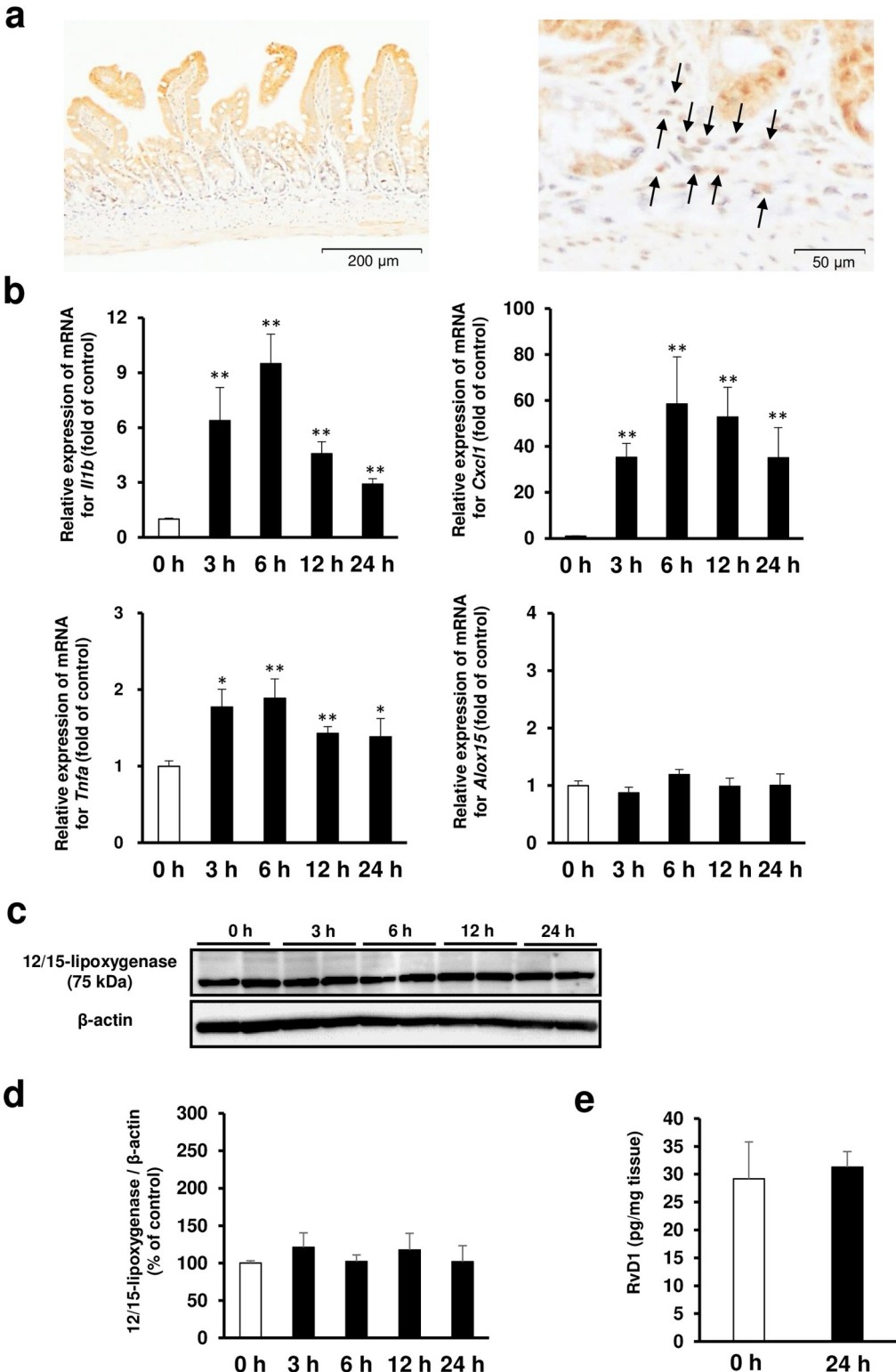

**Fig 2. Dynamics and localization of 12/15-lipoxygenase in small intestinal tissue.** (a) Representative image of immunohistochemical staining for localization analysis of 12/15-lipoxygenase protein in the small intestinal tissue. Bars in histological images: 200 μm. The figure on the right is a magnified view showing inflammatory cells with arrows. Bars in histological images: 50 μm. (b) Time course of gene expression levels of interleukin-1β (*Il1b*), CXCL1/ keratinocyte chemoattractant (*Cxcl1*), and tumor necrosis factor-α (*Tnfa*) and 12/15-lipoxygenase (*Alox15*) after

administration of indomethacin. $N$ = 5–8. $^{*}p < 0.05$, $^{**}p < 0.01$ vs. untreated control group (0 h group). (c) Representative images of western blotting analysis for time course of 12/15-lipoxygenase protein expression after administration of indomethacin. (d) Relative amount of normalized 12/15-lipoxygenase protein expressed as a percentage of untreated control (0 h group). $N$ = 4. (e) Concentration of resolvin D1 in the small intestinal tissue of untreated control mice (0 h group) and mice administered with indomethacin (24 h group). Small intestinal tissue samples were obtained from mice 24 h after indomethacin administration. N = 6. All data are from a single experiment, representative of at least two independent experiments and expressed as mean ± standard error of mean. $^{*}$ $p < 0.05$, $^{**}$ $p < 0.01$.

## Inhibition of 12/15-lipoxygenase by baicalein aggravates indomethacin-induced small intestinal damage

Pre-administration of baicalein, an inhibitor of the activity of 12/15-lipoxygenase, aggravated indomethacin-induced small intestinal damage. In particular, the lesion index increased by 1.6-fold compared with that of vehicle-treated mice (Fig 3A and 3B). Baicalein also promoted the expression of *Il1b*, *Tnfa*, and *Cxcl1* (Fig 3C). In addition, exogenous administration of resolvin D1 prevented the aggravation of intestinal damage by baicalein (Fig 3D). To corroborate the above experimental results, we confirmed that the administration of baicalein to the control group reduced the concentration of resolvin D1 in the small intestinal tissue in a dose-dependent manner (Fig 3E).

## Dynamics of ALX/FPR2 expression in small intestinal tissue

Similar to the expression of inflammatory cytokines, ALX/FPR2 protein expression peaked 6 to 12 h after administration of indomethacin (Fig 4A and 4B). Immunoreactivity against ALX/FPR2 was observed in intestinal epithelial cells and inflammatory cells of the lamina propria (Fig 4C). In particular, double staining for ALX/FPR2 with F4/80 showed that ALX/FPR2 was expressed in macrophages in the lamina propria (Fig 4D).

## Pharmacological inhibition of the resolvin D1 receptor ALX/FPR2 aggravates indomethacin-induced small intestinal damage

We further investigated the effects of Boc-1 and WRW4, inhibitors of the resolvin D1 receptor ALX/FPR2, on indomethacin-induced small intestinal damage. Exacerbation of mucosal lesions was observed after pretreatment with each of the inhibitors; specifically, administration of Boc-1 and WRW4 induced a 1.8-fold and 1.5-fold increase, respectively, in the lesion index (Fig 5A and 5B).

## Discussion

In the present study, we demonstrated that resolvin D1 exerts protective properties against NSAID-induced small intestinal damage. Exogenous supplementation of resolvin D1 protected the small intestine from damage induced by NSAIDs, while inhibition of 12/15-lipoxygenase, the enzyme responsible for the production of resolvin D1, exacerbated NSAID-induced small intestinal damage. Moreover, inhibition of ALX/FPR2, the receptor for resolvin D1, resulted in further deterioration of NSAID-induced small intestinal damage.

Accumulating evidence indicates that resolvin D1 is a key player in the resolution of inflammation [1]. One of the mechanisms through which resolvin D1 promotes resolution of inflammation is linked to its anti-inflammatory properties. In fact, resolvin D1 inhibits lipopolysaccharide (LPS)-induced transcriptional activation of TNF-α by modulating the nuclear factor κB (NF-κB) signaling pathway in human macrophages [21]. In addition, resolvin D1 downregulates the expression of IL-1β via inhibition of NOD-like receptor family

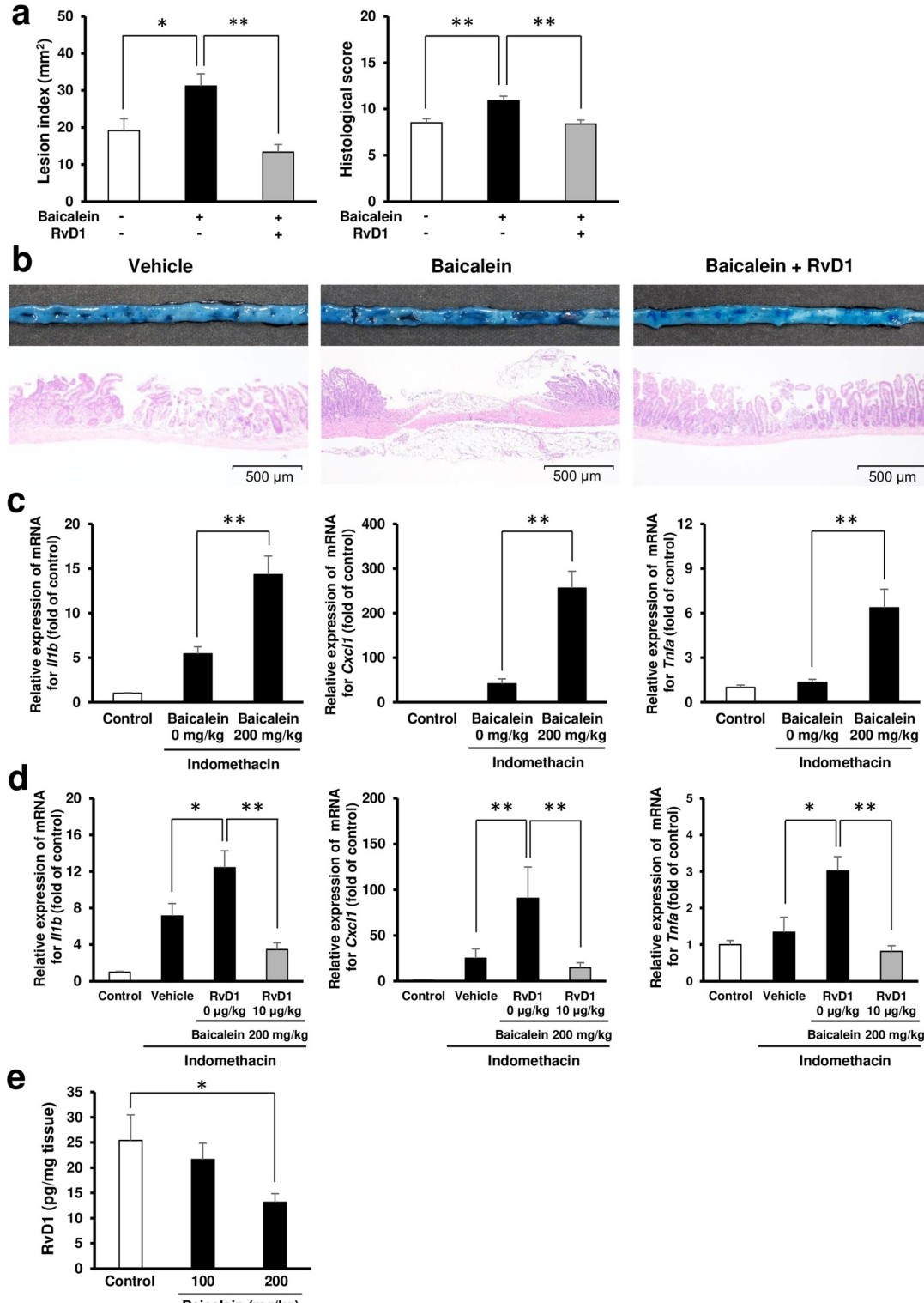

**Fig 3. Effect of inhibition of activity of 12/15-lipoxygenase on indomethacin-induced small intestinal damage.** (a) The effect of baicalein, a 12/15-lipoxygenase inhibitor, on indomethacin-induced small intestinal damage. Lesion index and histological score of the group pre-administered with baicalein and the group pre-administered baicalein (200 mg/kg BW) and resolvin D1 (RvD1: 10 μg/kg BW). $N$ = 7–8. Baicalein and resolvin D1 were administered 3 and 24 h before administration of indomethacin. (b) Representative macroscopic and microscopic images of small intestine 24 h after administration of

indomethacin in mice pre-administered with baicalein (200 mg/kg BW) and RvD1 (10 μg/kg BW). Bars in histological images: 500 μm. (c) The gene expression levels of interleukin-1β (*Il1b*), CXCL1/keratinocyte chemoattractant (*Cxcl1*), and tumor necrosis factor-α (*Tnfa*) 6 h after administration of indomethacin in mice pre-administered with baicalein (0, 200 mg/kg BW). The mRNA levels were determined by quantitative reverse transcription polymerase chain reaction and expressed as ratios relative to the mean value for small intestinal tissue from untreated control mice. $N$ = 6–8. (d) The gene expression levels of *Il1b*, *Cxcl1*, and *Tnfa* 6 h after administration of indomethacin (10 mg/kg BW) in mice pre-administered with baicalein (200 mg/kg BW) concomitant with RvD1 (0, 10 μg/kg BW). Baicalein and resolvin D1 were administered 3 and 24 h before administration of indomethacin. $N$ = 5–8. (e) Concentration of resolvin D1 in the small intestinal tissue of the group pre-administered with baicalein (100 and 200 mg/kg BW) and the control group. Baicalein was administered 24 and 48 h before sacrifice. $N$ = 6. All data are from a single experiment, representative of at least two independent experiments and expressed as mean ± standard error of mean. $^*$ $p < 0.05$, $^{**}$ $p < 0.01$.

proteins such as pyrin domain-1 containing 3 (NLRP3), thereby inhibiting the inflammasome pathway in rats with streptozotocin-induced diabetic retinopathy or hyperhomocysteinemia-induced podocyte injury [22, 23]. With regard to the pathophysiology of NSAID-induced small intestinal damage, we previously revealed that LPS and high-mobility group box 1 (HMGB1) liberated from damaged epithelial cells induce activation of Toll-like receptor (TLR) 4. Subsequently, TLR4 stimulates NF-κB to produce TNF-α through an MyD88-dependent pathway, resulting in NSAID-induced small intestinal damage [24, 25]. In addition, we have previously demonstrated that the NLRP3 inflammasome activates IL-1β via the TLR4 signaling pathway, which is a critical step for induction and promotion of NSAID-induced small intestinal damage [26]. Altogether, inhibition of the expression of TNF-α and IL-1β via inhibition of the TLR4- NF-κB and NLRP3 inflammasome signaling pathways could be proposed as the possible mechanism through which resolvin D1 ameliorates NSAID-induced small intestinal damage.

In the present study, we demonstrated that 12/15-lipoxygenase, a major synthetic enzyme for the production of resolvin D1 in mice, was mainly present in the small intestinal epithelium and inflammatory cells, consistent with a previous report [27]. The 12/15-lipoxygenase gene was constitutively expressed regardless of the time after indomethacin administration. We further showed that administration of baicalein, an inhibitor of the activity of 12/15-lipoxygenase, exacerbated NSAID-induced small intestinal damage and was accompanied by a reduced accumulation of endogenous resolvin D1 in small intestinal tissue; conversely, exogenous supplementation of resolvin D1 negated the deleterious effect of baicalein on NSAID-induced small intestinal damage. These results indicate that 12/15-lipoxygenase is constitutively expressed in the small intestinal mucosa to contribute to the regulation of inflammation via the production of resolvin D1. Consistent with the present study, the significance of 12/15-lipoxygenase in physiological homeostasis has been reported for a variety of pathologies in other organs. For example, 12/15-lipoxygenase contributes to skin homeostasis by regulating the infiltration of proinflammatory macrophages and proinflammatory signaling in dermal adipose tissue and in the dorsal skin [28]. Furthermore, deletion of 12/15-lipoxygenase accelerates the development of aging-associated and instability-induced osteoarthritis [29].

The contribution of 12/15-lipoxygenase-derived SPMs other than resolvin D1 to mucosal defense and repair against intestinal inflammation has also been reported. For instance, resolvin E1 exerts anti-inflammatory effect against experimental dextran sodium sulfate (DSS)-induced colitis and 2,4,6-trinitrobenzene sulfonic acid-induced colitis, and promoted intestinal wound healing; in addition, lipoxin A4 also exerts anti-inflammatory effect in a DSS-induced inflammatory bowel disease model and a ischemic/reperfusion injury model, counteracting LPS-induced acute intestinal inflammation [30–35]. With regard to NSAID-induced small intestinal damage, the present study suggests that the effect of 12/15-lipoxygenase on

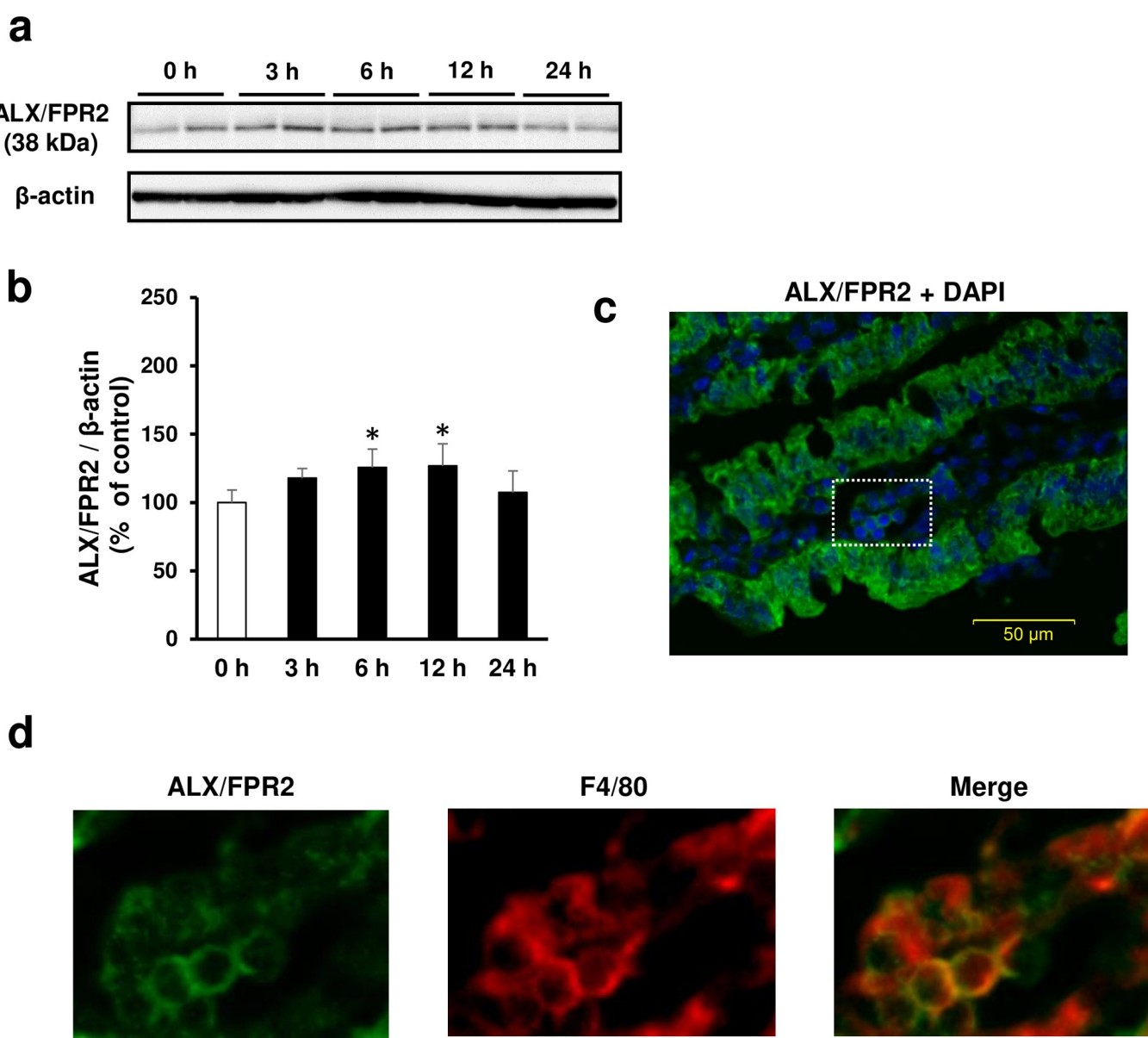

**Fig 4. Expression of resolvin D1 receptor, lipoxin A4 receptor /formyl peptide receptor 2 (ALX/FPR2) in the small intestine and its dynamics on indomethacin-induced small intestinal damage.** (a) Representative images of western blotting analysis for time course of ALX/FPR2 expression after administration of indomethacin (10 mg/kg BW). (b) Relative amount of normalized ALX/FPR2 protein expressed as a percentage of untreated control group (0 h group). *$p < 0.05$ vs. 0 h group. $N = 4$. (c) Representative image of immunofluorescent staining for localization analysis of ALX/FPR2 in damaged small intestinal mucosa. DAPI: 4',6-diamidino-2-phenylindole. Bars in histological images: 50 μm. (d) Expression of ALX/FPR2 and its colocalization with a marker of mature macrophages F4/80. These are enlarged views of the part indicated by white squares in image.

indomethacin-induced small intestinal damage is, at least in part, dependent on resolvin D1 because its exacerbation upon inhibition of 12/15-lipoxygenase was fully neutralized by exogenous supplementation of resolvin D1.

Recent studies have revealed that resolvin D1 exerts its anti-inflammatory effect via two G protein-coupled receptors, ALX/FPR2 and GPR32 [9]. Although the ortholog of human ALX/FPR2 is present in mice and rats, the murine ortholog of human GPR32 is absent [36]. ALX/FPR2 is expressed on various cells, including macrophages and intestinal epithelial cells, and

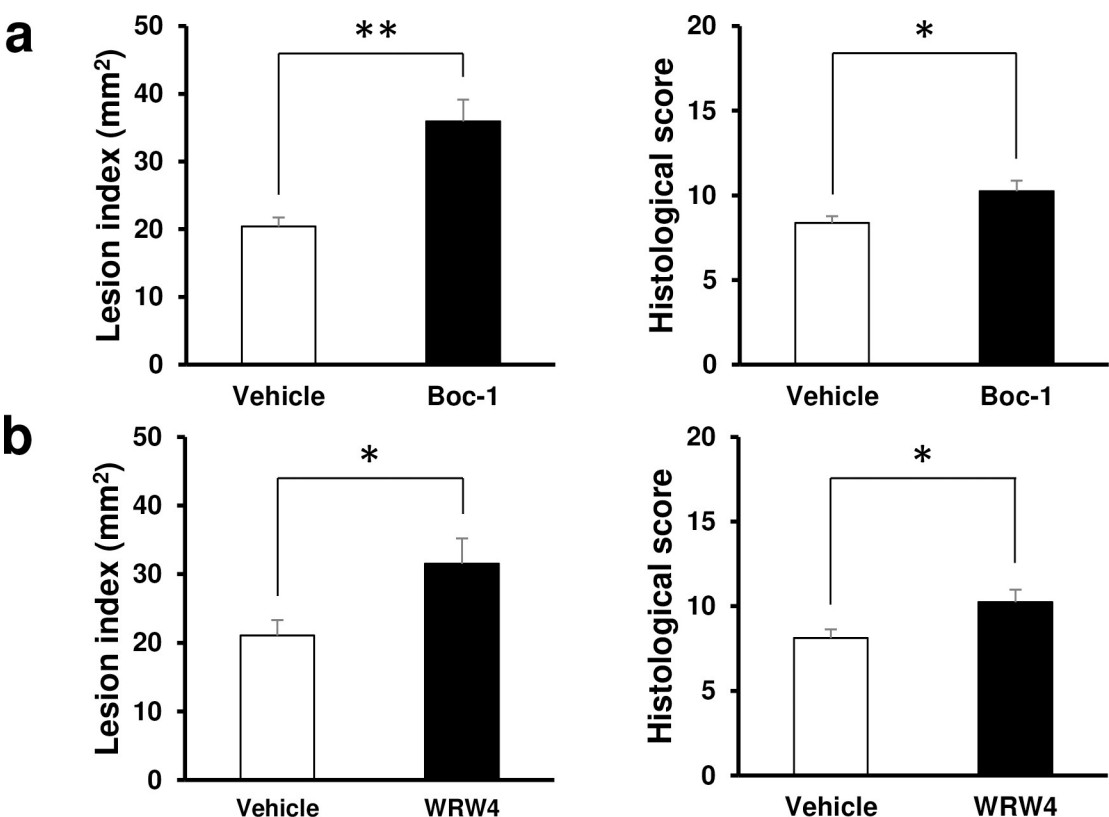

**Fig 5. Effect of inhibition of lipoxin A4 receptor /formyl peptide receptor 2 (ALX/FPR2) on indomethacin-induced small intestinal damage.** (a) Lesion index and histological score of small intestines 24 h after administration of indomethacin (10 mg/kg BW) in mice pre-administered with vehicle or ALX/FPR2 inhibitor Boc-1 (5 mg/kg BW). $N = 8$. (b) Lesion index and histological score of small intestines 24 h after administration of indomethacin (10 mg/kg BW) pre-administered with vehicle or ALX/FPR2 inhibitor WRW4 (1 mg/kg BW). $N = 6$–8. All data are from a single experiment, representative of at least two independent experiments and expressed as mean ± standard error of mean. * $p < 0.05$, ** $p < 0.01$.

recognizes resolvin D1, thereby promoting resolution of inflammation [9]. In the present study, we confirmed the expression of ALX/FPR2 in these cell types and found that the expression of this receptor peaked 6 to 12 h after indomethacin administration, similar to the expression of inflammatory cytokines. Such an increase in ALX/FPR2 expression was considered to be mainly because of indomethacin-promoted infiltration of inflammatory cells into the small intestinal tissue. Furthermore, indomethacin-induced small intestinal damage was aggravated by pre-administration of two different ALX/FPR2 inhibitors, Boc-1 and WRW4, indicating that these inhibitors blocked the anti-inflammatory effect of endogenous resolvin D1. Several reports have revealed the contribution of ALX/FPR2 in SPM-mediated resolution of inflammation *in vitro* and in animal models. For example, Bento and colleagues reported that the ALX/FPR2 inhibitor Boc-1 abolishes the anti-inflammatory effects of aspirin-triggered resolvin D1 on LPS-stimulated macrophages derived from murine bone marrow, thereby exacerbating murine DSS-induced colitis [37]. In addition, Gobbetti and colleagues reported that the blockade of ALX/FPR2 annuls its anti-inflammatory effects in an intestinal ischemia/reperfusion-induced injury mouse model [34]. Moreover, *Alx/Fpr2*-deficient mice show delayed mucosal restoration after DSS-induced colitis [38]. Consistently, Birkl and colleagues demonstrated that ALX/FPR2 regulates monocyte recruitment to promote intestinal mucosal wound repair [39]. In agreement with these studies, we also demonstrated that inhibition of ALX/

FPR2 abolished the inhibitory effect of endogenous resolvin D1 against indomethacin-induced small intestinal damage, leading to the exacerbation of such damage.

In conclusion, the present study demonstrated that 12/15-lipoxygenase-derived resolvin D1 contributes to mucoprotection against NSAID-induced small intestinal damage by exerting anti-inflammatory effects via activation of the ALX/FPR2 receptor.

## Supporting information

**S1 Table. The PCR primers and TaqMan probes used in the present study.**
(DOCX)

**S1 Fig. The original uncropped and unadjusted images of the western blotting analysis for 12/15 lipoxygenase.**
(TIF)

**S2 Fig. The original uncropped and unadjusted images of the western blotting analysis for ALX FRP2.**
(TIF)

## Acknowledgments

We thank Emi Yoshioka for providing technical assistance.

## Author Contributions

**Conceptualization:** Tetsuya Tanigawa.

**Data curation:** Tetsuya Tanigawa.

**Funding acquisition:** Tetsuya Tanigawa.

**Investigation:** Takuya Kuzumoto, Tetsuya Tanigawa.

**Methodology:** Takuya Kuzumoto, Tetsuya Tanigawa.

**Project administration:** Tetsuya Tanigawa.

**Supervision:** Tetsuya Tanigawa, Akira Higashimori, Hiroyuki Kitamura, Yuji Nadatani, Koji Otani, Shusei Fukunaga, Shuhei Hosomi, Fumio Tanaka, Noriko Kamata, Yasuaki Nagami, Koichi Taira, Toshio Watanabe, Yasuhiro Fujiwara.

**Validation:** Tetsuya Tanigawa, Akira Higashimori, Hiroyuki Kitamura, Yuji Nadatani, Koji Otani, Shusei Fukunaga, Shuhei Hosomi, Fumio Tanaka, Noriko Kamata, Yasuaki Nagami, Koichi Taira, Toshio Watanabe, Yasuhiro Fujiwara.

**Writing – original draft:** Takuya Kuzumoto.

**Writing – review & editing:** Tetsuya Tanigawa.

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
