## [Decision Letter · Decision Letter 0]

26 Jan 2021

PONE-D-21-00075

Protective role of resolvin D1, a pro-resolving lipid mediator, in nonsteroidal anti-inflammatory drug-induced small intestinal damage

PLOS ONE

Dear Dr. Tanigawa

Thank you for submitting your manuscript to PLOS ONE. After careful consideration, we feel that it has merit but does not fully meet PLOS ONE’s publication criteria as it currently stands. Therefore, we invite you to submit a revised version of the manuscript that addresses the points raised during the review process.

We look forward to receiving your revised manuscript.

Kind regards,

Hiroyasu Nakano, M.D., Ph.D.

Academic Editor

PLOS ONE

Journal Requirements:

2.At this time, we request that you  please report additional details in your Methods section regarding animal care, as per our editorial guidelines:

(1) Please include the method of euthanasia

(2) Please describe the operative care received by the animals, including the frequency of monitoring and the criteria used to assess animal health and well-being.

Thank you for your attention to these requests.

3. Please provide the product number and any lot numbers of the antibodies purchased for your study.

4.PLOS ONE now requires that authors provide the original uncropped and unadjusted images underlying all blot or gel results reported in a submission’s figures or Supporting Information files. This policy and the journal’s other requirements for blot/gel reporting and figure preparation are described in detail at https://journals.plos.org/plosone/s/figures#loc-blot-and-gel-reporting-requirements and https://journals.plos.org/plosone/s/figures#loc-preparing-figures-from-image-files. When you submit your revised manuscript, please ensure that your figures adhere fully to these guidelines and provide the original underlying images for all blot or gel data reported in your submission. See the following link for instructions on providing the original image data: https://journals.plos.org/plosone/s/figures#loc-original-images-for-blots-and-gels.

6.Thank you for submitting the above manuscript to PLOS ONE. During our internal evaluation of the manuscript, we found significant text overlap between your submission and the following previously published works, some of which you are an author.

- https://www.gastrojournal.org/article/S0016-5085(20)32578-6/pdf?referrer=https%3A%2F%2Fapi.ithenticate.com%2F

- https://www.intechopen.com/books/municipal-solid-waste-management/life-cycle-inventory-lci-modeling-of-municipal-solid-waste-msw-management-systems-in-kosodrza-commun

https://www.nature.com/articles/mi201589?code=4eb62728-cfde-4eba-97dd-898b4180c86b&error=cookies_not_supported

Please revise the manuscript to rephrase the duplicated text, cite your sources, and provide details as to how the current manuscript advances on previous work. Please note that further consideration is dependent on the submission of a manuscript that addresses these concerns about the overlap in text with published work.

Additional Editor Comments:

Although both reviewers feel the study is potentially interesting, one of the reviewers claims a flaw of the authors’ experimental conditions. Specifically, the reviewer strongly recommends the authors to collaborate with lipid biochemists to correctly measure lipid mediators including resolving D1 in their samples.

Reviewers' comments:

Reviewer's Responses to Questions

**Comments to the Author**

1. Is the manuscript technically sound, and do the data support the conclusions?

Reviewer #1: Partly

Reviewer #2: Yes

2. Has the statistical analysis been performed appropriately and rigorously? 

Reviewer #1: Yes

Reviewer #2: Yes

3. Have the authors made all data underlying the findings in their manuscript fully available?

Reviewer #1: Yes

Reviewer #2: Yes

4. Is the manuscript presented in an intelligible fashion and written in standard English?

Reviewer #1: Yes

Reviewer #2: Yes

5. Review Comments to the Author

Reviewer #1: Kuzumoto et al showed the resolving D1, an anti-inflammatory lipid mediator derived from DHA ameliorated the NSAIDs-induced intestinal damage by inhibiting the intestinal inflammation. An inhibitor for 12/15-lipoxygenase, a critical enzyme for resolving D1 synthesis, increased the expression of inflammatory cytokines and exacerbated the intestinal damage, which was again ameliorated by exogenous resolving D1. Most of the experiments and their results are nicely organized and discussed, except for the quantification of resolving D1 by Elisa.

Major comments

In figure 3a-d, they showed the marked increase in the lesion index, histological score and the expression of inflammatory cytokines by baicalein, a 12/15 lipoxygenase inhibitor, however, they showed the minimal reduction of resolving D1 by baicalein treatment in Figure 3e. This small change in resolving D1 can not explain the huge difference in the clinical scores and inflammation. This is possibly due to the inadequate measurement of resolving D1. They extracted fatty acid fraction from intestinal tissues by methanol and quantified resolving D1 using an Elisa kit, which does not give the reliable values. Currently, lipid mediators including SPM2 should be measured by mass-spectrometry as proposed by Bo Burla et al. in J. Lipid Res. 2018;59:2001-2017. The reviewer strongly recommends the authors to collaborate with lipid biochemists to correctly measure lipid mediators including resolving D1 in their samples.

Reviewer #2: In the manuscript entitled "Protective role of resolvin D1, a pro-resolving lipid mediator, in nonsteroidal anti-inflammatory drug-induced small intestinal damage" by Kuzumoto et al, the authors demonstrate that administration of resolvin D1 ameliorates NSAIDs-induced damages of the small intestine in mice (Fig. 1). The expression of the resolvin D1-producing enzyme (12/15-lipoxygenase) is not affected after administration of indomethacin (Fig. 2). The indomethacin-induced intestinal damages are aggravated in mice that are pre-treated with the inhibitor of 12/15-lipoxygenase, and the aggravation is abrogated by exogenous resolvin D1 (Fig. 3). The indomethacin-induced intestinal damages are also aggravated by the inhibitors of a resolvin D1 receptor (ALX1/FPR2), which is mainly expressed in macrophages (Fig. 4 and Fig. 5).

I feel that the present study provides important findings regarding the role of the endogenous lipid mediator resolvin D1 in regulation of inflammation, and thus seems to be suitable for publication if the following minor issues are adequately addressed.

Specific comments:

1) The authors should measure the concentration of resolvin D1 in the small intestine before and after administration of indomethacin, while they have shown that the expression of 12/15-lipolyxegease is not affected by the administration. Alternatively, appropriate references should be cited in terms of the alteration of the enzymatic activity during the inflammatory responses.

2) The scale bar should be included in the right panel of Fig. 2a.

3) In Figs. 1b and 1e, the time point at the observation after indomethacin administration should be presented (possibly 24 h).

4) For all experiments in Fig. 3, the timing and duration of pre-treatment of baicalein should be provided.

5) The gene names should be formally written for Il1b and Tnf (both in italic), instead of those in the current manuscript.

6. PLOS authors have the option to publish the peer review history of their article (what does this mean?). If published, this will include your full peer review and any attached files.

Reviewer #1: No

Reviewer #2: No

---

## [Author Response · Author response to Decision Letter 0]

27 Mar 2021

Response to the editors and reviewers

PONE-D-21-00075

Protective role of resolvin D1, a pro-resolving lipid mediator, in nonsteroidal anti-inflammatory drug-induced small intestinal damage

Editor’s comments

1.“Please ensure that your manuscript meets PLOS ONE's style requirements, including those for file naming.”

>>>We ensured that our manuscript meets PLOS ONE’s style requirements, including those for file naming.

2. “At this time, we request that you please report additional details in your Methods section regarding animal care, as per our editorial guidelines:

(1) Please include the method of euthanasia

(2) Please describe the operative care received by the animals, including the frequency of monitoring and the criteria used to assess animal health and well-being.”

>>We really appreciate the editor’s advice. In accordance with the editors’ advice, we reported the details of the method of euthanasia and operative care in the revised manuscript. 

3. “Please provide the product number and any lot numbers of the antibodies purchased for your study”.

>>>In the revised manuscript, we provided the product number and the lot numbers of the antibodies purchased for our study. 

4. “PLOS ONE now requires that authors provide the original uncropped and unadjusted images underlying all blot or gel results reported in a submission’s figures or Supporting Information files”. 

>>>We provided the original uncropped and unadjusted images underlying all blot results reported in a Supporting Information file. 

5. “Please include captions for your Supporting Information files at the end of your manuscript, and update any in-text citations to match accordingly”. 

>>> We included captions for your Supporting Information files at the end of our revised manuscript.

6. “Thank you for submitting the above manuscript to PLOS ONE. During our internal evaluation of the manuscript, we found significant text overlap between your submission and the following previously published works, some of which you are an author.

- https://www.gastrojournal.org/article/S0016-5085(20)32578-6/pdf?referrer=https%3A%2F%2Fapi.ithenticate.com%2F

- https://www.intechopen.com/books/municipal-solid-waste-management/life-cycle-inventory-lci-modeling-of-municipal-solid-waste-msw-management-systems-in-kosodrza-commun

https://www.nature.com/articles/mi201589?code=4eb62728-cfde-4eba-97dd-898b4180c86b&error=cookies_not_supported

Please revise the manuscript to rephrase the duplicated text, cite your sources, and provide details as to how the current manuscript advances on previous work. Please note that further consideration is dependent on the submission of a manuscript that addresses these concerns about the overlap in text with published work”.

>>>We revised the manuscript carefully to rephrase the duplicated text in accordance with the results of the present study. 

Additional Editor Comments:

“Although both reviewers feel the study is potentially interesting, one of the reviewers claims a flaw of the authors’ experimental conditions. Specifically, the reviewer strongly recommends the authors to collaborate with lipid biochemists to correctly measure lipid mediators including resolvin D1 in their samples.”

>>>Please see “the Response to Reviewer #1’s Comment”. 

Comments to the Author

Response to Reviewer #1: 

Major comments

“In figure 3a-d, they showed the marked increase in the lesion index, histological score and the expression of inflammatory cytokines by baicalein, a 12/15 lipoxygenase inhibitor, however, they showed the minimal reduction of resolvin D1 by baicalein treatment in Figure 3e. This small change in resolvin D1 can not explain the huge difference in the clinical scores and inflammation. This is possibly due to the inadequate measurement of resolvin D1. They extracted fatty acid fraction from intestinal tissues by methanol and quantified resolvin D1 using an Elisa kit, which does not give the reliable values. Currently, lipid mediators including SPM2 should be measured by mass-spectrometry as proposed by Bo Burla et al. in J. Lipid Res. 2018;59:2001-2017. The reviewer strongly recommends the authors to collaborate with lipid biochemists to correctly measure lipid mediators including resolvin D1 in their samples”.

>>>We really appreciate the reviewer’s advice. In accordance with the reviewer’s advice, we tried to measure the concentration of resolvin D1 in the small intestinal tissue with help of the expert of mass-spectrometory analysis at the Research Support Platform of Osaka City University Graduate School of Medicine. However, the result was not reasonable. At the moment, for our study, the EIA assay provided by Cayman Chemical is the most reliable method, and this assay kit is used successfully in the several studies including the following papers; 

Chen YC, Su MC, Chin CH, Lin IC, Hsu PY, Liou CW, Huang KT, Wang TY, Lin YY, Zheng YX, Hsiao CC, Lin MC. Formyl peptide receptor 1 up-regulation and formyl peptide receptor 2/3 down-regulation of blood immune cells along with defective lipoxin A4/resolvin D1 production in obstructive sleep apnea patients. PLoS One. 2019 May 22;14(5):e0216607. doi: 10.1371/journal.pone.0216607. PMID: 31116781; PMCID: PMC6530856.

Parashar K, Schulte F, Hardt M, Baker OJ. Sex-mediated elevation of the specialized pro-resolving lipid mediator levels in a Sjögren's syndrome mouse model. FASEB J. 2020 Jun;34(6):7733-7744. doi: 10.1096/fj.201902196R. Epub 2020 Apr 11. PMID: 32277856.

Nordgren TM, Friemel TD, Heires AJ, Poole JA, Wyatt TA, Romberger DJ. The omega-3 fatty acid docosahexaenoic acid attenuates organic dust-induced airway inflammation. Nutrients. 2014 Nov 27;6(12):5434-52. doi: 10.3390/nu6125434. PMID: 25436433; PMCID: PMC4276977.

Zhang L, Terrando N, Xu ZZ, Bang S, Jordt SE, Maixner W, Serhan CN, Ji RR. Distinct Analgesic Actions of DHA and DHA-Derived Specialized Pro-Resolving Mediators on Post-operative Pain After Bone Fracture in Mice. Front Pharmacol. 2018 May 1;9:412. doi: 10.3389/fphar.2018.00412. PMID: 29765320; PMCID: PMC5938385.

To confirm the validation of the assay system using this EIA kit in the present study, and to response the comment suggested by Reviewer #2 about concentration of resolvin D1, we performed the additional experiment. In the revised manuscript, we have demonstrated the following results. 

1. There was no difference in concentration of resolvin D1 in the small intestine between the vehicle-administered control group and indomethacin-administered group (Fig 2e). 

2. The inhibitory effect of baicalein on concentration of resolvin D1 in small intestinal tissue is dose-dependent (Fig 3e). 

These results support the validation of the methods of measurement of concentration of resolvin D1 in small intestinal tissue by using the EIA assay. In the revised manuscript, we added the results of the additional experiments. 

Response to Reviewer #2: 

1) “The authors should measure the concentration of resolvin D1 in the small intestine before and after administration of indomethacin, while they have shown that the expression of 12/15-lipolyxegease is not affected by the administration. Alternatively, appropriate references should be cited in terms of the alteration of the enzymatic activity during the inflammatory responses.”

>>>We really appreciate the reviewer’s suggestion. In accordance with the reviewer’s advice, we measured the concentration of resolvin D1 in the small intestine before and after administration of indomethacin. In consistent with the constant expression of 12/15-lipoxygenase, there was no difference in concentration of resolvin D1 in the small intestine between the vehicle-administered control group and indomethacin-administered group (Fig 2e). We added the data in the revised manuscript. 

2) “The scale bar should be included in the right panel of Fig. 2a.”

>>>We added the scale bar in the right panel of Fig.2a. 

3) “In Figs. 1b and 1e, the time point at the observation after indomethacin administration should be presented (possibly 24 h).”

>>>We added the information of the time point at the observation after indomethacin administration in the Figure legends of the revised manuscript. 

4) “For all experiments in Fig. 3, the timing and duration of pre-treatment of baicalein should be provided.”

>>>We provided the information about the timing and duration of pre-treatment of baicalein in the Figure legends of the revised manuscript. 

5) “The gene names should be formally written for Il1b and Tnf (both in italic), instead of those in the current manuscript.”

>>>We corrected the gene names in the revised manuscript.

---

## [Decision Letter · Decision Letter 1]

15 Apr 2021

Protective role of resolvin D1, a pro-resolving lipid mediator, in nonsteroidal anti-inflammatory drug-induced small intestinal damage

PONE-D-21-00075R1

Dear Dr. Tanigawa

We’re pleased to inform you that your manuscript has been judged scientifically suitable for publication and will be formally accepted for publication once it meets all outstanding technical requirements.

Although one reviewer still has a concern about the quantification of the indicated lipids of mass spectrometry, I have agreed with the authors’ response that the quantification did not work well at least under the authors’ experimental conditions. Thus, I have accepted the manuscript.

Kind regards,

Hiroyasu Nakano, M.D., Ph.D.

Academic Editor

PLOS ONE

Additional Editor Comments (optional):

Reviewers' comments:

Reviewer's Responses to Questions

**Comments to the Author**

1. If the authors have adequately addressed your comments raised in a previous round of review and you feel that this manuscript is now acceptable for publication, you may indicate that here to bypass the “Comments to the Author” section, enter your conflict of interest statement in the “Confidential to Editor” section, and submit your "Accept" recommendation.

Reviewer #1: (No Response)

Reviewer #2: All comments have been addressed

2. Is the manuscript technically sound, and do the data support the conclusions?

Reviewer #1: Yes

Reviewer #2: (No Response)

3. Has the statistical analysis been performed appropriately and rigorously? 

Reviewer #1: Yes

Reviewer #2: (No Response)

4. Have the authors made all data underlying the findings in their manuscript fully available?

Reviewer #1: Yes

Reviewer #2: (No Response)

5. Is the manuscript presented in an intelligible fashion and written in standard English?

Reviewer #1: Yes

Reviewer #2: (No Response)

6. Review Comments to the Author

Reviewer #1: As a lipid biochemist, the reviewer can not accept the results drawn by SPM measurements by ELISA, even though some papers using this ELISA have been published.

Reviewer #2: (No Response)

7. PLOS authors have the option to publish the peer review history of their article (what does this mean?). If published, this will include your full peer review and any attached files.

Reviewer #1: No

Reviewer #2: No

---

## [Editor Report · Acceptance letter]

20 Apr 2021

PONE-D-21-00075R1 

Protective role of resolvin D1, a pro-resolving lipid mediator, in nonsteroidal anti-inflammatory drug-induced small intestinal damage 

Dear Dr. Tanigawa:

I'm pleased to inform you that your manuscript has been deemed suitable for publication in PLOS ONE. Congratulations! Your manuscript is now with our production department. 

Kind regards, 

on behalf of

Professor Hiroyasu Nakano 

Academic Editor

PLOS ONE